# The associations between *Schistosoma mansoni* infection, pre-treatment symptoms, praziquantel side effects, and treatment efficacy in Ugandan school-aged children

Huanghehui Yu[1,2], Moses Arinaitwe[1,3], Adriko Moses[3], Narcis Kabatereine[3], Edridah M. Tukahebwa[3], David W. Oguttu[3], Aidah Wamboko[3], Annet Namukuta[3], Annet Enzaru[3], Joaquin M. Prada[4], Alan Fenwick[5], Joanne P. Webster[5,6], Poppy H. L. Lamberton[1,5]*, Jessica Clark[1]*

1 School of Biodiversity, One Health and Veterinary Medicine, College of Medical, Veterinary and Life Sciences, University of Glasgow, Glasgow, United Kingdom, 2 Section of Paediatric Infectious Disease, Department of Infectious Disease, Imperial College London, London, United Kingdom, 3 Vector Borne & NTD Control Division, Ministry of Health, Kampala, Uganda, 4 School of Veterinary Medicine, University of Surrey, Guildford, United Kingdom, 5 Department of Infectious Disease Epidemiology, Imperial College London, London, United Kingdom, 6 Department of Pathobiology and Population Sciences, Royal Veterinary College, Hawkshead, United Kingdom

* poppy.lamberton@glasgow.ac.uk (PHLL); jessica.clark@glasgow.ac.uk (JC)

## Abstract

### Background

Over 240 million people have schistosomiasis. Mass drug administration (MDA) with the anthelmintic praziquantel is the cornerstone of control. Treatment side effects are commonly observed and may be associated with dying worms. Side effects have also been reported as a reason for reduced MDA uptake, potentially resulting in those most in need refusing repeated treatment. An improved understanding of the association between side effects and infection intensity, pre-treatment health, and drug efficacy, may help inform education campaigns to facilitate increased MDA uptake.

### Methods

Using latent class analyses, Beta regression, and dose response curves, we analysed egg and antigen parasitological data alongside health and side-effects survey data pre- and post-praziquantel-treatment from two primary schools (Bugoto Lake View (LV) and Musubi Church of God (CoG)) in *Schistosoma mansoni* high endemicity Ugandan villages to understand whether pre-treatment infection status or intensity were related to 1) pre-treatment symptoms, 2) post-treatment side effects, and 3) whether parasite clearance after treatment was associated with side effects.

**Data availability statement:** All anonymised data and code to reproduce the analysis are available at https://github.com/iamjessclark/symptomsSideEffects.git.

**Funding:** Data collection was funded by The Bill & Melinda Gates Foundation 36202 through the Schistosomiasis Control Initiative (Now Unlimit Health), and a Medical Research Council Studentship (both supporting PHLL). PHLL and HY salaries were funded by European Research Council Starting Grant EPX0270821. JC salary is funded by NERC NE/W003333/1. The funders had no role in study design, data collection and analysis, decision to publish, or preparation of the manuscript.

**Competing interests:** The authors have declared that no competing interests exist.

## Principal findings

At Bugoto LV: Abdominal pain, blood-in-stool, and itching/rash symptoms were non-linearly associated with infection intensity; Diarrhoea, headache and vomiting side effects were non-linearly associated with infection intensity. At Musubi CoG: Blood-in-stool, headache, and pain-when-urinating symptoms were non-linearly associated with infection intensity; Abdominal pain, diarrhoea, and vomiting side effects were non-linearly associated with infection intensity. There was no relationship between infection status (infected/ uninfected) and symptoms or side effects at either school. No association was found between infection clearance and the presence of side effects at either school.

## Conclusions

We show no evidence that being infected predisposes someone to side effects, nor that side effects are related to treatment efficacy. Relationships between pre-treatment infection intensity and pre-treatment symptoms or post-treatment side effects varied by school suggesting unmeasured factors such as co-infections or other health conditions could impact symptom and side effect reporting.

### Author summary

Schistosomiasis is a neglected tropical disease, widely treated by mass drug administration with praziquantel, but side effects often occur, and are also reported as a reason for refusing treatment. This study examined the impact of praziquantel treatment for intestinal schistosomiasis at two schools in Uganda, focusing on the relationships between pre-treatment infection intensity, pre-treatment symptoms, post-treatment side effects, and treatment efficacy. Our analyses showed that relationships between symptoms or side effects and infection intensity are not linear, and were most reported in those who did not have the highest infection intensities. The symptoms and side effects reported were not consistent across schools. Furthermore, the occurrence of side effects did not correlate with the probability of successful infection clearance, or with the probability of being infected pre-treatment. Our findings highlight the complex nature of praziquantel induced side effects. Further studies are needed to understand how and why symptoms and/or side effects vary, even between demographically and geographically similar communities as well as across individuals. These insights could guide the development of health messaging, which may improve treatment uptake, therefore improving overall coverage of mass drug administration programs, which is vital to achieving the World Health Organization's 2030 goal for schistosomiasis, of elimination as a public health problem.

## Introduction

Schistosomiasis is a neglected tropical disease (NTD) with more than 240 million people infected across 78 countries [1–3]. It is prevalent in tropical and sub-tropical regions and disproportionately affects people living in resource-limited settings, with inadequate access to sanitation and safe water. Infection occurs through contact with contaminated freshwater, when the human infective stage of the lifecycle, cercariae, penetrates the skin. In the case of *Schistosoma mansoni*, a major causative agent of intestinal schistosomiasis, adult female and male worms sexually reproduce, and each pair can produce between 100 and 300 eggs each day [4]. Eggs are excreted in the stool and hatch into miracidia once they contact freshwater. Miracidia infect *Biomphalaria* snails, where they asexually reproduce shedding thousands of cercariae, completing the life-cycle.

Adult *S. mansoni* worms reside within the mesenteric venules surrounding the bowels, and cannot be directly observed *in situ*. As a result, there is no gold standard diagnostic such that all diagnostic methods serve as proxies for true infection status. There are two World Health Organization (WHO) recommended diagnostics for *S. mansoni* [3]. Kato-Katz thick smears, which count eggs in stool, are highly specific and give an estimate of infection intensity, but egg shedding is variable between samples and days, and they lack sensitivity especially in low intensity infections or post treatment, often underestimating prevalence, and overestimating drug efficacy [5]. The point-of-care circulating cathodic antigen test (POC-CCA) detects antigens excreted in host urine, that have been regurgitated by feeding worms. They are more sensitive than Kato-Katz, but lack specificity and are only semi-quantitative [6]. Latent class models enable improved insights from imperfect diagnostics [7–9] addressing many of these challenges.

Whilst Kato-Katz and POC-CCAs can indicate infection presence and intensity, they do not inform about infection induced morbidity. Eggs that are excreted in the stool are important indicators of ongoing transmission, but a proportion of eggs remain in the body and it is these that cause the majority of the morbidity [10]. Acute schistosomiasis is generally associated with a fever and other symptoms collectively known as Katayama Fever [11]. Untreated infections often lead to chronic schistosomiasis, associated with impaired cognitive potential, granuloma formation, hyperaemia, anaemia, oedema, colonic ulceration and stunted growth [12–14]. Mild chronic infections can cause symptoms such as abdominal pains, blood-in-stool or -urine, diarrhoea, dysuria, and dysentery [4,15]. In prolonged and heavy infections, more severe illnesses can develop, including hepatosplenomegaly with up to 200,000 deaths a year attributed to schistosomiasis [16].

Mass drug administration (MDA) with the anthelmintic praziquantel has been the mainstay of control efforts [17–19]. Referring to the distribution of treatment to a target population regardless of the recipient's infection status, MDA is recommended by the WHO for the control of six NTDs including schistosomiasis [17]. Praziquantel is the main drug approved for the treatment of schistosomiasis [20,21]. A standard dose of 40 mg/kg is recommended, but treatment efficacy can vary greatly [22–27]. Additionally, treatment is ineffective against juvenile worms (those less than approximately seven-weeks-old) [28–31], which likely contributes to the rapid resurgence often seen post-treatment [27,32], especially in high endemicity areas where exposure can be constant, such that juvenile worms and new, post-treatment exposures, can be common and rapid. Praziquantel damages the tegument of *S. mansoni*, leading to rapid depolarization, muscle contraction, and subsequent death of the adult worms [31]. It is hypothesised that a spike in praziquantel-induced adult worm mortality can cause a rush of antigens to be released, such that individuals who report more side effects may have had high numbers of dying or damaged adult worms, and so be more likely to have cleared their infections [33]. Common side effects of praziquantel that have been reported include abdominal pain/stomach discomfort, diarrhoea, dizziness, headache, nausea and vomiting [14]. Clinical observation suggest that 30% to 60% of patients experience side effects after taking praziquantel, but they are usually mild and resolve within 24 hours [34–36]. Side effects can lead to patients to refusing treatment in subsequent rounds [36,37], particularly if the recipient knows they are infected and so assumes the side effects will be more intense [38]. However, there is little evidence to suggest that presence or intensity of symptoms or side effects are connected to pre-treatment infection presence or intensity [39,40]. Furthermore, due to the fact that

adult worms cannot be directly observed, it remains unclear whether the presence of side effects reported by an individual are indicative of individual-level treatment efficacy.

As of 2021, schistosomiasis has been targeted by the WHO for elimination as a public-health problem (EPHP) by 2030 (validated when <1% of school-aged children have a heavy infection intensity, classified as >400 eggs per gram (epg) of stool using one Kato-Katz for *S. mansoni*). As MDA remains the cornerstone of achieving this ambitious target, it is necessary to understand the interaction between infection prevalence and intensity, symptoms, side effects, and treatment efficacy, so that evidence-based information can be given to communities regarding the risk of side effects and what they might expect as a result of treatment. Here we leverage a latent class analysis framework [7,27] fit to egg- and antigen-based epidemiological diagnostic data to infer the unobservable, 'true', infection status, thereby enabling a more robust estimate of true clearance after treatment. We then use the model outputs to investigate how pre-treatment symptoms and post-treatment side effects relate to infection dynamics and treatment efficacy. Specifically, we address five main hypotheses: 1) people with *S. mansoni* infections are more likely to report infection-associated pre-treatment symptoms (referred to as symptoms here after) than those who are uninfected; 2) these symptoms are more likely in people with higher pre-treatment infection intensities in comparison to uninfected people or those with lower infection intensities (referred to as infection intensities here after); 3) people with *S. mansoni* infections are more likely to report post-treatment side effects when treated with praziquantel than uninfected people (referred to as side effects here after); 4) these side effects are more likely in people with heavier infections in comparison to those uninfected or with lower infection intensities when treated with praziquantel; and 5) people reporting more side effects are more likely to have successfully cleared their infections (assuming that side effects are caused by damaged *S. mansoni* adult worms – though we do not test this causal hypothesis directly).

## Methodology

### Ethics statement

Ethical approvals were gained from the Uganda National Council for Science and Technology (Memorandum of Understanding: sections 1.4, 1.5, and 1.6) and the Imperial College Research Ethics Committee (EC NO: 03.36. R&D No: 03/SB/033E). Due to low parental literacy, informed parental consent was recorded by teachers during school committee meetings prior to the start of the study. Teachers and community leaders verbally consented to the study happening in their community. Each head teacher gave written permission for the pupils to participate in the study. Before being included in the study, each participant also verbally assented and understood that they could leave the study at any time with no impact on whether they received their MDA treatment.

### Study sites & recruitment

Data were collected in 2004 from school-aged children at two schools: Bugoto Lake View (LV) and Musubi Church of God (CoG) primary schools, in Mayuge District, Uganda. Bugoto and Musubi villages are based on the shores of Lake Victoria and are highly endemic for *S. mansoni*. Children at Bugoto LV were first recruited in 2003 by the Schistosomiasis Control Initiative and 123 students successfully re-recruited in this study and 68 students were first recruited in 2004 from Musubi CoG (Table 1). A total of 162 participants had data successfully collected pre- and post-treatment for analysis, with 94 from Bugoto LV and 68 from Musubi CoG (S1 Table). In 2004, Musubi CoG had never received praziquantel treatment. In Bugoto LV, MDA had started one year previously, in 2003. All participants were given unique IDs, and sex and age were recorded.

### Diagnostic methods

Kato-Katz thick smears were used to quantify *S. mansoni* infection intensities. In both schools, a stool sample was collected once a day for three consecutive days for every participant at each sampling point, and from each stool sample

**Table 1. Summary statistics of raw data from Bugoto Lake View (LV) and Musubi Church of God (CoG) primary schools in 2004. Sym: pre-treatment symptom; SE: post-treatment side effect. Pre-treatment infection intensity categories are classified based on WHO guidelines [3]. Percentages in the 'Bugoto LV' and 'Musubi CoG' columns reflect within-school proportions (i.e., the number of children with the symptom divided by the sample count number assessed in that school). Percentages in the 'Total' column reflect overall proportions across both schools combined.**

| | Schools | | Total |
|---|---|---|---|
| | **Bugoto LV** | **Musubi CoG** | |
| **Sample count (n)** | 94 | 68 | 162 |
| **Sex** | | | |
| Female: | 49 (52.1%) | 34 (50.0%) | 83 (51.2%) |
| Male: | 45 (47.9%) | 34 (50.0%) | 79 (48.8%) |
| **Infection prevalence** | 77 (81.9%) | 62 (91.2%) | 139 (85.8%) |
| **Infection intensity (median [IQR], EPG)** | 96.0 [26.0 – 248.0] | 204.0 [83.0 – 563.5] | 124.0 [35.2 – 374.4] |
| **Infection intensity categories** | | | |
| Light (1–99 EPG): | 37 (39.4%) | 15 (22.1%) | 52 (32.1%) |
| Moderate (100–399 EPG): | 28 (29.8%) | 25 (36.8%) | 53 (32.7%) |
| Heavy (>400 EPG): | 12 (12.8%) | 22 (32.4%) | 34 (21.0%) |
| **POC-CCA categories** | | | |
| Pre-treatment: | | | |
| Negative | 8 (8.5%) | 8 (11.7%) | 16 (9.9%) |
| Trace | 11 (11.7%) | 3 (4.4%) | 14 (8.6%) |
| + | 17 (18.1%) | 23 (33.8%) | 40 (24.7%) |
| ++ | 28 (29.8%) | 24 (35.3%) | 52 (34.2%) |
| +++ | 4 (4.3%) | 7 (10.3%) | 11 (6.8%) |
| Four-Weeks Post-treatment: | | | |
| Negative | 30 (31.9%) | 32 (47.1%) | 62 (38.3%) |
| Trace | 2 (2.1%) | 11 (16.2%) | 13 (8.0%) |
| + | 12 (12.8%) | 21 (30.9%) | 33 (20.4%) |
| ++ | 1 (1.1%) | 1 (1.5%) | 2 (1.2%) |
| +++ | 2 (2.1%) | 0 (0.0%) | 2 (1.2%) |
| **Abdominal pain** | | | |
| Sym: | 51 (54.3%) | 59 (86.8%) | 110 (67.9%) |
| SE: | 52 (55.3%) | 51 (75.0%) | 103 (63.6%) |
| **Blood-in-stool** | | | |
| Sym: | 23 (24.5%) | 28 (41.2%) | 51 (31.5%) |
| SE: | 0 (0.0%) | 0 (0.0%) | 0 (0.0%) |
| **Diarrhoea** | | | |
| Sym: | 35 (37.2%) | 55 (80.9%) | 90 (55.6%) |
| SE: | 33 (35.1%) | 33 (48.5%) | 66 (40.7%) |
| **Headache** | | | |
| Sym: | 41 (43.6%) | 59 (86.8%) | 100 (61.7%) |
| SE: | 22 (22.0%) | 23 (33.8%) | 45 (27.8%) |
| **Itching/rash** | | | |
| Sym: | 30 (31.9%) | 37 (54.4%) | 67 (41.4%) |
| SE: | 0 (0.0%) | 0 (0.0%) | 0 (0.0%) |
| **Nausea** | | | |
| Sym: | 37 (39.4%) | 53 (77.9%) | 90 (55.6%) |
| SE: | 0 (0.0%) | 0 (0.0%) | 0 (0.0%) |

*(Continued)*

**Table 1.** (Continued)

| | Schools | | Total |
|---|---|---|---|
| | **Bugoto LV** | **Musubi CoG** | |
| **Pain-when-urinating** | | | |
| Sym: | 26 (27.7%) | 29 (42.7%) | 55 (34.0%) |
| SE: | 0 (0.0%) | 0 (0.0%) | 0 (0.0%) |

duplicate Kato-Katz thick smears were produced, resulting in up to six slides per participant per timepoint. Eggs identified as *S. mansoni* were counted under a microscope, averaged across the slides and multiplied by 24 to calculate epg of stool [41]. One urine sample was also collected per participant per timepoint and a single POC-CCA test was conducted at the time of collection on fresh urine samples using the original commercial POC-CCA test with 2 drops of buffer added to the urine, with results graded as Negative, Trace, +, ++, or +++ depending on the lateral flow assay's coloured response. After three days of samples had been collected at baseline, all participants were given bread and juice, prior to observed praziquantel treatment at a dose of 40 mg/kg, regardless of infection status [42] as per the MDA programme for schistosomiasis in Uganda, but using scales for weight rather than the dose pole. Three days of duplicate Kato Katz were repeated at one- and four-weeks post-treatment and one day of POC-CCA at four weeks (as drug efficacy protocols were still being established at this stage). Participants with *S. mansoni* egg counts of >100 epg at one-week post-treatment were administered a second round of praziquantel for ethical reasons (36.2% of participants in Bugoto LV and 42.6% in Musubi CoG). S4 Fig shows re-treatment did not strongly influence the probability of clearance and therefore we did not split the analyses, using pre-treatment and four-week post-treatment data for hypotheses 5.

## Symptoms and side-effect surveys

Pre-treatment, each participant answered a health survey about a range of symptoms including abdominal pain, blood-in-stool, diarrhoea, headache, nausea, pain-when-urinating, and/or rash/itching and if they had ever had them. The survey was developed for use in both intestinal and urinary schistosomiasis areas, and the complete absence of *Schistosoma haematobium* in this area was still being confirmed and although not expected to be associated with intestinal schistosomiasis, pain-when-urinating was reported pre-treatment and therefore kept in the analyses. It has also been significantly associated with reduced health-related quality of life in individuals with and without *S. mansoni* evidencing a complicated relationship between pain-when-urinating and intestinal schistosomiasis that is not well understood [43]. Twenty-four hours post-treatment all participants were asked about any side effects they might have experienced, including abdominal pain, blood-in-stool, diarrhoea, difficulty breathing, feeling dizzy, headache, nausea, pain-when-urinating, itching/rash, face swelling, vomiting, vomiting blood, and/or weakness. Both the symptoms and side effects were based on self reporting, except if vomiting was observed after treatment, in which case it was noted down and the participant was retreated. If they vomited again they were referred to the health facility and excluded from the analyses, but not the wider study or MDA programme. The proportions of children reporting symptoms and side effects from Bugoto LV in S2 Fig, and Musubi CoG are shown in S3 Fig.

## Data handling & statistical tools

Analyses and visualisations were conducted in R v4.1.2 [44]. Data manipulation and visualisations were undertaken with the *tidyverse* package [45]. Analysis of residuals for frequentist mixed effects models was conducted using the *DHARMa* package [46]. Due to significant differences between the schools in the symptoms and side effects reported, models were run and results reported individually by school.

## Hidden Markov model (HMM)

We fitted an existing HMM framework to the Kato-Katz and POC-CCA data, as imprecise indicators of the latent infection state, to estimate true infection status pre- and post-treatment and subsequently the probability of clearance post-treatment [7,27]. We used raw repeated egg counts, and POC-CCA scores that were converted such that they ranged from 0 to 4 (equating to Negative (0), Trace (1), + (2), ++ (3), +++ (4)). Further model details are described elsewhere [7,27], but in short, the model predicted the true unobservable egg intensity for each individual drawn from a gamma-distributed population-level mean at baseline for each school, with an autoregressive random walk component for subsequent timepoints. To account for overdispersion in individual egg counts, the likelihood on the Kato-Katz egg counts was drawn from a negative binomial distribution. The likelihood for the POC-CCA data was derived from a logistic function, where the numerator was the maximum possible numerical score (in this case +++ represented as 4), and the real number of the denominator, the true unobservable infection intensity for that individual as estimated by the model. This assumes that as true infection intensity increases, so does the POC-CCA score, without making any *a priori* assumption regarding the interpretation of POC-CCA scores (including trace readings) as positive or negative. Instead, we used the POC-CCA scores to reflect a range of infection intensities and probabilities of infection. We calculated the infection probability for each individual at each of the three timepoints from the estimated latent infection status, over 40,000 iterations with two chains. The population-level clearance and reinfection proportions at each timepoint were drawn from a weakly informative Beta distribution. This estimated proportion was then used to at each timepoint to estimate whether an individual $i$ had cleared infection or become reinfected at each timepoint $t$ as a 0 or 1, drawn from a Bernoulli distribution. In turn, the clearance and reinfection probabilities for each individual at each timepoint could be calculated as the proportion of iterations in which the model predicted true clearance for each individual at time point $t$ out of all 40,000 iterations. The *runjags* package [47] was used to fit the model.

## Hypotheses 1: Association between infection probability and symptoms

The probability that an individual was infected pre-treatment was used as the explanatory variable, and the symptom reported by participants before treatment (binary variables) was used as the response variable. This was modelled with a generalised linear model with Beta distributed errors by *glmmTMB* package in R [48]. A model was run for each symptom (abdominal pain, blood in stool, diarrhoea, headache, itching/rash, nausea, and pain when urinating). The decision to run separate models rather than one model with all symptoms was due to the issue of collinearity. For example, abdominal pain and diarrhoea are not necessarily independent of one another, but equally can occur exclusively. The same logic is applied in the following tests where relevant.

## Hypotheses 2: Association between infection intensity and symptoms

We randomly sampled 5,000 iterations from the posterior distribution of estimated egg counts for each person. For each iteration, we calculated proportions of egg counts and symptoms and fitted a dose-response model using the *drc* package in R [49]. The response variable was the proportion of participants that reported the symptom. The explanatory variables were the egg count categories, where each category contained 120 epg egg counts (e.g., 1–120, 121–240; 0 eggs as estimated by the model therefore truly no eggs, was considered as a separate category). In Musubi CoG, egg count categories ranged from 0 to 3,000 epg, and, in Bugoto LV, they ranged from 0 to 1680 epg. From the resulting model fits, we extracted the 2.5%, 50%, and 97.5% quantiles of proportion of participants that reported the symptom to summarise uncertainty. We used the median of all estimated egg counts across the 40,000 iterations from the HMM model to fit a single dose-response mode. This fit represents the average association between egg count and symptom reporting, while the corresponding figure shows the variability around this relationship based on the posterior sampling approach.

## Hypotheses 3: Association between infection probability and side effects

The analysis was identical to hypothesis 1, but with side effects as the response variables.

### Hypotheses 4: Association between infection intensity and side effects

Analysis was identical to hypothesis 2, but the proportion of participants in each egg count category that reported side effects was used instead.

### Hypotheses 5: Association between side effects and the probability of clearance

To assess whether there was an association between side effects and the probability that an individual cleared infection after treatment we used a sequence of generalised linear models with beta error distribution, using the *glmmTMB* package [48], to find whether any of the side effects (abdominal pain, headache, diarrhoea, and vomiting) explained the probability of clearance. The response variable was the probability of clearance obtained from the HMM for each participant, transformed to enable the Beta distribution to accommodate boundary values [0,1] [50]. The independent variable was the presence of each side effect separately. Estimates were obtained for each school.

## Results

### Summary statistics

Musubi CoG exhibited the highest median and widest interquartile range (204.0 epg [83.0 – 563.5]) in infection intensities and the highest prevalence (91.2%) compared to Bugoto LV (81.9%) (Fig 1, Table 1).

The most frequently reported symptom was abdominal pain, with a prevalence of 67.9%. This was followed by headache, reported by 61.7%. Specifically, in Musubi CoG, 86.8% of participants reported headaches, and an equal proportion reported abdominal pain, compared to 43.6% and 54.3% in Bugoto LV, respectively. In terms of side effects, abdominal pain was the most commonly reported, with a 63.6% overall prevalence across both schools. Blood-in-stool, itching and/or rashing, nausea, and pain-when-urinating were not reported as side effects in Bugoto LV or Musubi CoG.

### Hypotheses 1: Association between infection probability and symptoms

Abdominal pain, blood-in-stool, diarrhoea, headache, itching/rash, nausea, and pain-when-urinating were reported as symptoms in both schools. A high proportion of children had a high probability of infection (posterior probability > 0.9), with 93.6% in Bugoto LV and 92.6% in Musubi CoG. None of the symptoms were related to the probability of being infected with *S. mansoni* in either of the schools (Fig 1 and Table 2).

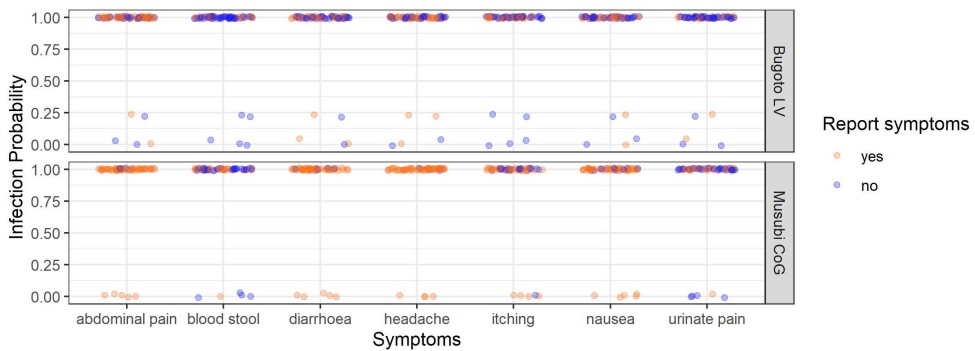

**Fig 1. The *Schistosoma mansoni* infection probability (from the latent class analysis) of every participant in Bugoto Lake View (LV) (top graph) and Musubi Church of God (CoG) (bottom graph) primary schools who reported (orange), or did not report (blue) pre-treatment symptoms (x-axis).** Note the jittering of points to prevent them overlaying one another.

**Table 2. Regression results of the *Beta* model of *Schistosoma mansoni* infection probability (from the latent class analysis) with each symptom as a fixed-effect variable. The intercept was the group of participants who did not report the symptom for each symptom in Bugoto Lake View (LV) and Musubi Church of God (CoG) primary schools respectively.**

| School | Symptoms | Effect | Estimate ± SD | p-value# |
|---|---|---|---|---|
| **Bugoto LV** | | | | |
| | Abdominal pain | (Intercept) | -0.574 ± 1.014 | 0.571 |
| | | Infection probability | 1.523 ± 1.056 | 0.149 |
| | Blood-in-stool | (Intercept) | -21.02 ± 2021.30 | 0.992 |
| | | Infection probability | 20.35 ± 2021.30 | 0.992 |
| | Diarrhoea | (Intercept) | 0.475 ± 1.008 | 0.637 |
| | | Infection probability | -0.594 ± 1.043 | 0.569 |
| | Headache | (Intercept) | 0.264 ± 0.994 | 0.790 |
| | | Infection probability | -0.017 ± 1.029 | 0.986 |
| | Itching/rash | (Intercept) | -21.14 ± 2016.92 | 0.992 |
| | | Infection probability | 20.94 ± 2016.92 | 0.992 |
| | Nausea | (Intercept) | -0.489 ± 1.010 | 0.629 |
| | | Infection probability | 0.550 ± 1.044 | 0.599 |
| | Pain-when-urinating | (Intercept) | -0.469 ± 1.013 | 0.643 |
| | | Infection probability | -0.085 ± 1.051 | 0.936 |
| **Musubi CoG** | | | | |
| | Abdominal pain | (Intercept) | 18.64 ± 2930.36 | 0.995 |
| | | Infection probability | -16.26 ± 2930.36 | 0.996 |
| | Blood-in-stool | (Intercept) | -1.386 ± 1.121 | 0.216 |
| | | Infection probability | 1.281 ± 1.152 | 0.266 |
| | Diarrhoea | (Intercept) | 18.64 ± 2930.37 | 0.995 |
| | | Infection probability | -16.52 ± 2930.37 | 0.996 |
| | Headache | (Intercept) | 18.57 ± 3262.67 | 0.995 |
| | | Infection probability | -15.66 ± 3262.67 | 0.996 |
| | Itching/rash | (Intercept) | 1.385 ± 1.121 | 0.217 |
| | | Infection probability | -0.833 ± 1.158 | 0.472 |
| | Nausea | (Intercept) | 17.64 ± 1777.33 | 0.992 |
| | | Infection probability | -15.97 ± 1777.33 | 0.993 |
| | Pain-when-urinating | (Intercept) | -1.412 ± 1.130 | 0.211 |
| | | Infection probability | 1.378 ± 1.161 | 0.235 |

# Significance codes: 0 – 0.001 (***); 0.001 – 0.01 (**); 0.01 – 0.05 (*); > 0.05 (non-significance).

## Hypotheses 2: Association between infection intensity and symptoms

Abdominal pain, blood-in-stool, diarrhoea, headache, itching/rash, nausea, and pain-when-urinating were reported as symptoms in both schools. The dose-response curves for Bugoto LV (Fig 2, left, pink lines) indicated that increasing *S. mansoni* infection intensity was associated with a significant decline in the reporting of diarrhoea and headache. There is initially almost no relationship between infection intensity and nausea, or pain-when-urinating, but a significant negative relationship is driven by the decline observed in those with 481–600 epg or more. Alternatively, reports of abdominal pain, blood-in-stool, and itching/rash display a non-linear relationship with infection intensity, initially showing a positive relationship with increasing infection intensity up to a peak (varies by symptom), before then decreasing with infection intensity (Fig 2, Table 3). Similarly, in Musubi CoG (Fig 2, right, pink lines) reports of abdominal pain, diarrhoea, itching/rash and nausea show little relationship with infection intensity to a point, at which reports declined significantly with rising infection intensity, driven by

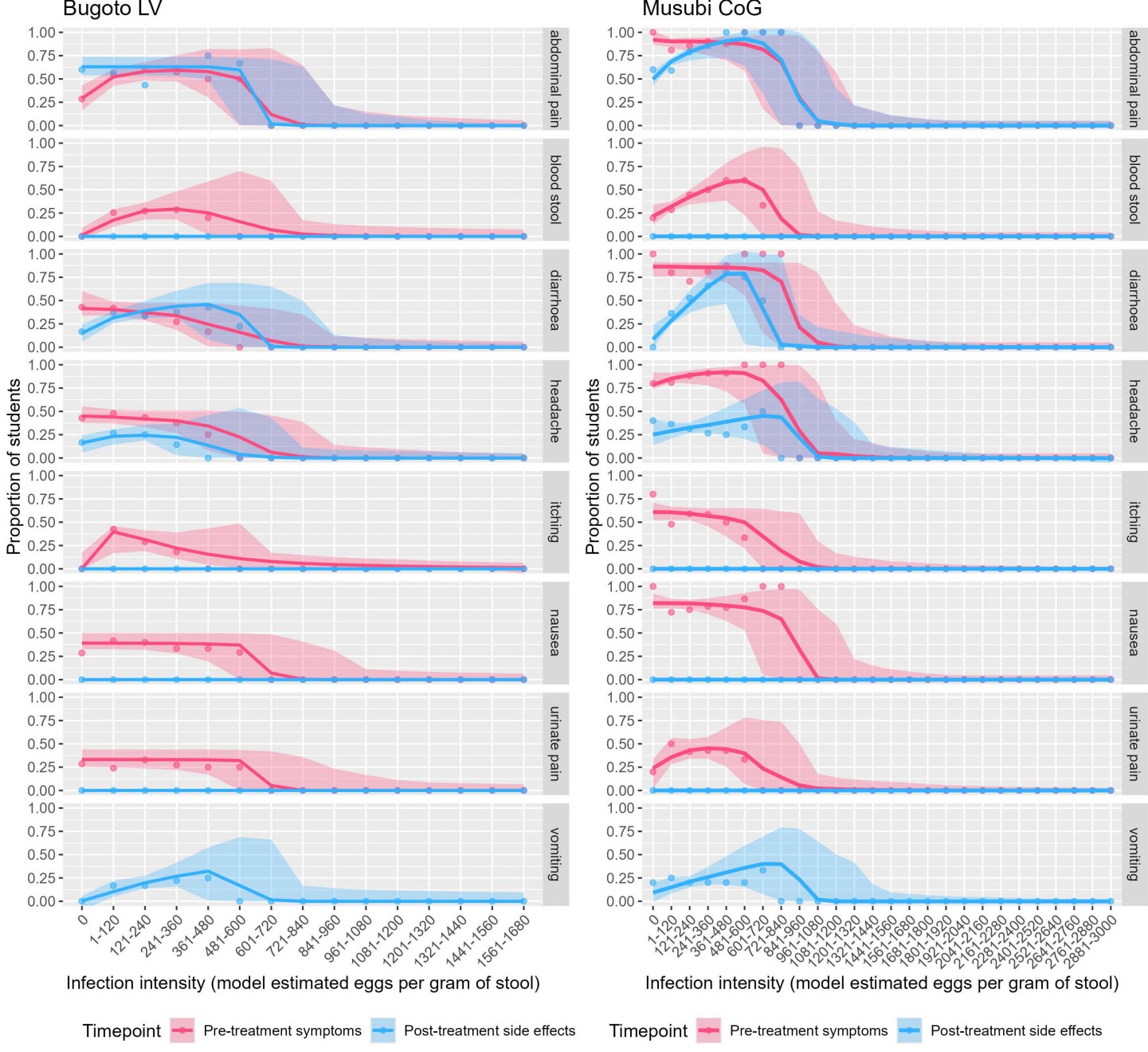

**Fig 2. *Schistosoma mansoni* infection intensity dose response curves (from the latent class analysis) for the corresponding pre-treatment symptoms and post-treatment side effects of *Schistosoma mansoni* infection, including abdominal pain, blood-in-stool, diarrhoea, headache, itching and rash, nausea, and pain-when-urinating, for participants from Bugoto Lake View (LV) and Musubi Church of God (CoG) primary schools.**

small samples sizes in the highest egg count categories. The frequency of blood-in-stool, headache, and pain-when-urinating also displayed non-linear relationships with infection intensity, with a significant positive relationship peaking before declining once more (Fig 2, Table 3). The distribution of infection intensities by each symptom are provided in S1 Fig.

**Table 3.** *Schistosoma mansoni* infection intensity dose response results for the pre-treatment symptoms. The intercept was the group of participants not reporting the symptom in Bugoto Lake View (LV) and Musubi Church of God (CoG) primary schools. Statistical significance is indicated as follows: *** p<0.001, ** p<0.01, * p<0.05, and p≥0.05 is considered non-significant.

| School | Symptoms | Effect | Estimate±SE | p-value# |
|---|---|---|---|---|
| **Bugoto LV** | **Abdominal pain** | Generalised Cedergreen-Ritz-Streibig (hormesis) | | |
| | | Slope | 46.469±9.095 | **< 0.001 (***)** |
| | | Lower limit | -0.001±0.004 | 0.9 |
| | | Upper limit | 0.552±0.009 | **< 0.001 (***)** |
| | | ED50 | 6.298±0.079 | **< 0.001 (***)** |
| | | Hormesis | -0.724±0.054 | **< 0.001 (***)** |
| | | Shape parameter | -6.477±10 | 0.525 |
| | **Blood-in-stool** | Generalised Cedergreen-Ritz-Streibig (hormesis) | | |
| | | Slope | 22.075±1.197 | **< 0.001 (***)** |
| | | Lower limit | 0±0.001 | 0.668 |
| | | Upper limit | 0.281±0.003 | **< 0.001 (***)** |
| | | ED50 | 5.199±0.016 | **< 0.001 (***)** |
| | | Hormesis | -0.763±0.011 | **< 0.001 (***)** |
| | | Shape parameter | -1.66±0.06 | **< 0.001 (***)** |
| | **Diarrhoea** | Brain-Cousens (hormesis) with lower limit fixed at 0 | | |
| | | Slope | 25.955±3.611 | **< 0.001 (***)** |
| | | Lower limit | 0 | |
| | | Upper limit | 0.564±0.005 | **< 0.001 (***)** |
| | | ED50 | 5.297±0.052 | **< 0.001 (***)** |
| | | Hormesis | -0.072±0.002 | **< 0.001 (***)** |
| | **Headache** | Weibull with lower limit at 0 | | |
| | | Slope | 8.124±0.902 | **< 0.001 (***)** |
| | | Lower limit | 0 | |
| | | Upper limit | 0.442±0.008 | **< 0.001 (***)** |
| | | ED50 | 5.265±0.047 | **< 0.001 (***)** |
| | **Itching/rash** | Generalised Cedergreen-Ritz-Streibig (hormesis) | | |
| | | Slope | 6.679±0.849 | **< 0.001 (***)** |
| | | Lower limit | -0.001±0.003 | 0.777 |
| | | Upper limit | 0.412±0.02 | **< 0.001 (***)** |
| | | ED50 | 3.641±0.118 | **< 0.001 (***)** |
| | | Hormesis | -1.119±0.068 | **< 0.001 (***)** |
| | | Shape parameter | -3.514±5.097 | 0.499 |
| | **Nausea** | Weibull with lower limit at 0 | | |
| | | Slope | 27.668±10.39 | **0.014 (**)** |
| | | Lower limit | 0 | |
| | | Upper limit | 0.354±0.01 | **< 0.001 (***)** |
| | | ED50 | 6.616±0.199 | **< 0.001 (***)** |
| | **Pain-when-urinating** | Weibull with lower limit at 0 | | |
| | | Slope | 23.882±6.566 | **0.001 (***)** |
| | | Lower limit | 0 | |
| | | Upper limit | 0.277±0.007 | **< 0.001 (***)** |
| | | ED50 | 6.579±0.142 | **< 0.001 (***)** |

*(Continued)*

**Table 3.** (Continued)

| School | Symptoms | Effect | Estimate±SE | p-value# |
|---|---|---|---|---|
| **Musubi CoG** | **Abdominal pain** | Brain-Cousens (hormesis) with lower limit fixed at 0 | | |
| | | Slope | 82.313±39.068 | **0.047 (*)** |
| | | Lower limit | 0 | |
| | | Upper limit | 0.852±0.031 | **< 0.001 (***)** |
| | | ED50 | 8.509±0.236 | **< 0.001 (***)** |
| | | Hormesis | 0.017±0.006 | **0.013 (**)** |
| | **Blood-in-stool** | Brain-Cousens (hormesis) with lower limit fixed at 0 | | |
| | | Slope | 42.085±18.653 | **0.034 (*)** |
| | | Lower limit | 0 | |
| | | Upper limit | 0.214±0.021 | **< 0.001 (***)** |
| | | ED50 | 7.159±0.075 | **< 0.001 (***)** |
| | | Hormesis | 0.069±0.005 | **< 0.001 (***)** |
| | **Diarrhoea** | Log-logistic (ED50 as parameter) with lower limit at 0 | | |
| | | Slope | 98.527±75.708 | 0.206 |
| | | Lower limit | 0 | |
| | | Upper limit | 0.896±0.023 | **< 0.001 (***)** |
| | | ED50 | 8.592±0.369 | **< 0.001 (***)** |
| | **Headache** | Brain-Cousens (hormesis) with lower limit fixed at 0 | | |
| | | Slope | 68.154±16.427 | **< 0.001 (***)** |
| | | Lower limit | 0 | |
| | | Upper limit | 0.778±0.013 | **< 0.001 (***)** |
| | | ED50 | 8.329±0.084 | **< 0.001 (***)** |
| | | Hormesis | 0.036±0.003 | **< 0.001 (***)** |
| | **Itching/rash** | Brain-Cousens (hormesis) with lower limit fixed at 0 | | |
| | | Slope | 14.669±3.831 | **< 0.001 (***)** |
| | | Lower limit | 0 | |
| | | Upper limit | 0.635±0.033 | **< 0.001 (***)** |
| | | ED50 | 6.784±0.186 | **< 0.001 (***)** |
| | | Hormesis | -0.025±0.011 | **0.035 (*)** |
| | **Nausea** | Log-logistic (ED50 as parameter) with lower limit at 0 | | |
| | | Slope | 93.827±59.265 | 0.127 |
| | | Lower limit | 0 | |
| | | Upper limit | 0.859±0.024 | **< 0.001 (***)** |
| | | ED50 | 8.604±0.298 | **< 0.001 (***)** |
| | **Pain-when-urinating** | Generalised Cedergreen-Ritz-Streibig (hormesis) | | |
| | | Slope | 36.929±16.109 | **0.033 (*)** |
| | | Lower limit | -0.001±0.003 | 0.878 |
| | | Upper limit | 0.447±0.007 | **< 0.001 (***)** |
| | | ED50 | 6.177±0.082 | **< 0.001 (***)** |
| | | Hormesis | -0.763±0.043 | **< 0.001 (***)** |
| | | Shape parameter | -4.97±19430.231 | 1 |

#Significance codes: 0 − 0.001 (***); 0.001 − 0.01 (**); 0.01 − 0.05 (*); >0.05 (non-significance).

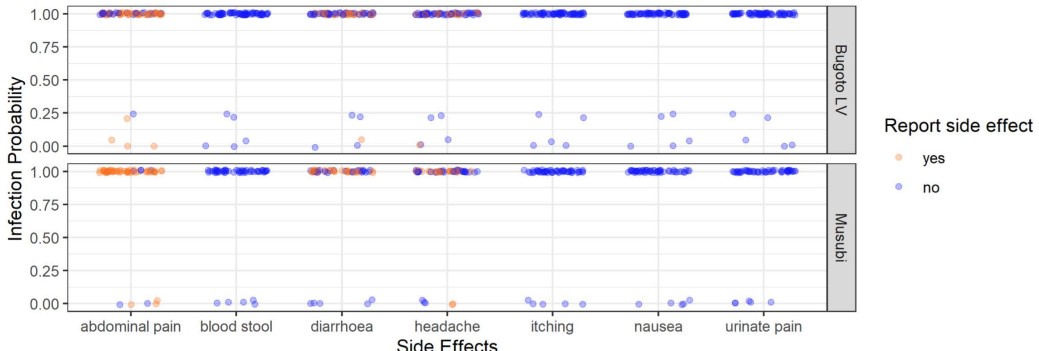

**Fig 3.** The *Schistosoma mansoni* infection probability (from the latent class analysis) of every participant in Bugoto Lake View (LV) (top graph) and Musubi Church of God (CoG) (bottom graph) primary schools who reported (orange), or did not report (blue), post-treatment side effects (x-axis). Note the jittering of points to prevent them overlaying one another.

### Hypotheses 3: Association between infection probability and side effects

There was no statistically significant association between *S. mansoni* infection probability and the presence of any side effects in both schools (Fig 3, Table 5). However, no participants reported blood-in-stool, itching/rash, nausea, or pain-when-urinating as side effects, whereas some participants did report them as symptoms (Fig 1).

### Hypotheses 4: Association between infection intensity and side effects

Blood-in-stool, itching/rash, nausea, and pain-when-urinating were not reported as side effects, irrespective of infection intensity, in either location (Fig 1, blue). In Bugoto LV abdominal pain, diarrhoea, headache, and vomiting were reported. Similar to the analysis on symptoms, abdominal pain was largely unrelated to infection intensity but a significant negative relationship was likely driven by so few people in the higher egg count categories (Fig 2, Table 4). The relationship with diarrhoea, headache, and vomiting was significant but non-linear and peaked within the 121–480 epg range (equivalent to WHO moderate to heavy infection intensities [3]), before declining to zero at higher intensities (Fig 2, Table 4). In Musubi CoG, there was a negligible relationship observed between infection intensity and headache but a significant negative relationship in the analysis was likely also driven by the small sample sizes in the highest egg count categories. A significant non-linear relationship was seen for abdominal pain, diarrhoea, and vomiting, where the prevalence peaked in the 481–600 epg range (equivalent to WHO heavy infection intensities [3]) before also decreasing to zero at the highest intensities (Fig 2, Table 4). The distribution of infection intensities by side effects is shown in S1 Fig.

### Hypotheses 5: Association between side effects and the probability of clearance

Clearance after treatment varied between the two schools (Fig 4) with the mean clearance probability estimated at 82.94% in Bugoto LV and 60.86% in Musubi CoG. The Mann-Whitney U test (W = 2967, $p = 0.424$) indicated that this difference was not statistically significant, suggesting similar distributions in clearance rates between the two schools, despite children at Bugoto LV having received treatment in the previous year and Musubi CoG being praziquantel naïve. At both schools, at the individual-level there was no association between whether an individual successfully cleared infection and whether they reported abdominal pain, diarrhoea, headaches, or vomiting as side effects (Table 6, Fig 4).

## Discussion

Praziquantel is administered through MDA to millions of people every year for schistosomiasis morbidity and transmission control, with the aim of EPHP by 2030. Despite the extent of treatment, coverage remains low in many places, and side

**Table 4.** *Schistosoma mansoni* infection intensity dose response results (from the latent class analysis) for side effects where dose response curves could be fitted. The intercept was the group of participants who did not reporting a side effect in every side effect in Bugoto Lake View (LV) and Musubi Church of God (CoG) primary schools. Statistical significance is indicated as follows: *** $p < 0.001$, ** $p < 0.01$, * $p < 0.05$, and $p \geq 0.05$ is considered non-significant.

| School | Side Effects | Effect | Estimate ± SE | p-value# |
|---|---|---|---|---|
| **Bugoto LV** | **Abdominal pain** | Weibull with lower limit at 0 | | |
| | | Slope | 34.833 ± 12.914 | **0.013 (**)** |
| | | Lower limit | 0 | |
| | | Upper limit | 0.616 ± 0.021 | **< 0.001 (***)** |
| | | ED50 | 6.734 ± 0.143 | **< 0.001 (***)** |
| | **Diarrhoea** | Generalised Cedergreen-Ritz-Streibig (hormesis) | | |
| | | Slope | 36.604 ± 20.246 | 0.086 |
| | | Lower limit | 0 ± 0.003 | 0.91 |
| | | Upper limit | 0.374 ± 0.008 | **< 0.001 (***)** |
| | | ED50 | 6.116 ± 0.072 | **< 0.001 (***)** |
| | | Hormesis | -0.563 ± 0.045 | **< 0.001 (***)** |
| | | Shape parameter | -2.337 ± 1.366 | 0.103 |
| | **Headache** | Brain-Cousens (hormesis) with lower limit fixed at 0 | | |
| | | Slope | 5.443 ± 0.57 | **< 0.001 (***)** |
| | | Lower limit | 0 | |
| | | Upper limit | 0.078 ± 0.037 | **0.048 (*)** |
| | | ED50 | 3.59 ± 0.243 | **< 0.001 (***)** |
| | | Hormesis | 0.093 ± 0.026 | **0.002 (**)** |
| | **Vomiting** | Generalised Cedergreen-Ritz-Streibig (hormesis) | | |
| | | Slope | 30.648 ± 4.828 | **< 0.001 (***)** |
| | | Lower limit | 0 ± 0.003 | 0.911 |
| | | Upper limit | 0.253 ± 0.04 | **< 0.001 (***)** |
| | | ED50 | 5.526 ± 0.085 | **< 0.001 (***)** |
| | | Hormesis | -0.671 ± 0.101 | **< 0.001 (***)** |
| | | Shape parameter | -0.816 ± 0.245 | **0.003 (**)** |
| **Musubi CoG** | **Abdominal pain** | Cedergreen-Ritz-Streibig with lower limit 0 | | |
| | | Slope | 74.228 ± 25.33 | **0.008 (**)** |
| | | Lower limit | 0 | |
| | | Upper limit | 0.021 ± 0.057 | 0.714 |
| | | ED50 | 8.47 ± 0.17 | **< 0.001 (***)** |
| | | Hormesis | 1.116 ± 0.076 | **< 0.001 (***)** |
| | **Diarrhoea** | Generalised Cedergreen-Ritz-Streibig (hormesis) | | |
| | | Slope | 34.025 ± 11.313 | **0.007 (**)** |
| | | Lower limit | -0.001 ± 0.005 | 0.846 |
| | | Upper limit | 1.009 ± 0.12 | **< 0.001 (***)** |
| | | ED50 | 7.082 ± 0.032 | **< 0.001 (***)** |
| | | Hormesis | -2.744 ± 0.312 | **< 0.001 (***)** |
| | | Shape parameter | -0.522 ± 0.085 | **< 0.001 (***)** |
| | **Headache** | Weibull with lower limit at 0 | | |
| | | Slope | 39.052 ± 9.889 | **< 0.001 (***)** |
| | | Lower limit | 0 | |
| | | Upper limit | 0.325 ± 0.01 | **< 0.001 (***)** |
| | | ED50 | 7.758 ± 0.098 | **< 0.001 (***)** |

*(Continued)*

**Table 4.** (Continued)

| School | Side Effects | Effect | Estimate ± SE | p-value# |
|---|---|---|---|---|
| | **Vomiting** | Brain-Cousens (hormesis) with lower limit fixed at 0 | | |
| | | Slope | 55.577 ± 15.896 | **0.002 (**)** |
| | | Lower limit | 0 | |
| | | Upper limit | 0.155 ± 0.02 | **< 0.001 (***)** |
| | | ED50 | 7.578 ± 0.14 | **< 0.001 (***)** |
| | | Hormesis | 0.018 ± 0.004 | **< 0.001 (***)** |

#Significance codes: 0 − 0.001 (***); 0.001 − 0.01 (**); 0.01 − 0.05 (*); >0.05 (non-significance).

**Table 5.** Regression results of the beta model of *Schistosoma mansoni* infection probability (from the latent class analysis) with each side effect as a fixed-effect variable. The intercept was the group of participants who not reporting the side effect in Bugoto Lake View (LV) and Musubi Church of God (CoG) primary schools respectively.

| School | Side Effects | Effect | Estimate ± SD | p-value# |
|---|---|---|---|---|
| **Bugoto LV** | | | | |
| | Abdominal pain | (Intercept) | 1.669 ± 1.317 | 0.205 |
| | | Infection probability | -0.9425 ± 1.346 | 0.484 |
| | Diarrhoea | (Intercept) | -1.415 ± 1.209 | 0.242 |
| | | Infection probability | 1.213 ± 1.237 | 0.327 |
| | Headache | (Intercept) | -1.292 ± 1.187 | 0.277 |
| | | Infection probability | 0.417 ± 1.221 | 0.733 |
| **Musubi CoG** | | | | |
| | Abdominal pain | (Intercept) | 0.392 ± 0.916 | 0..669 |
| | | Infection probability | 1.872 ± 1.031 | 0.069 |
| | Diarrhoea | (Intercept) | -17.65 ± 1777.31 | 0.992 |
| | | Infection probability | 18.15 ± 1777.31 | 0.992 |
| | Headache | (Intercept) | -0.398 ± 0.916 | 0.664 |
| | | Infection probability | 0.024 ± 0.959 | 0.980 |

# Significance codes: 0 − 0.001 (***); 0.001 − 0.01 (**); 0.01 − 0.05 (*); >0.05 (non-significance).

effects are reported as reasons for low uptake [37]. In this study we tested the relationship between *S. mansoni* infection prevalence, intensity, and pre-treatment self reported symptoms and post-treatment self reported side effects. We also tested whether side effects were related to drug efficacy four weeks after praziquantel. Our analyses do not entirely support previously published relationships [34,51] that suggest the severity of praziquantel treatment side effects are positively correlated with infection intensity of *S. mansoni.* Instead, we show that this relationship is variable across settings and non-linear, with side effects often peaking at moderate egg counts. Our results also do did not support the hypothesis that side effects are related to subsequent parasite clearance probability. Separately, despite Musubi CoG being drug-naïve prior to this study and Bugoto LV having received one prior round of MDA, praziquantel efficacy was comparable between the two schools. Overall, our results indicate that interactions between infection status, symptoms, side effects and treatment efficacy are complex making it difficult to predict individual responses to praziquantel treatment with certainty.

## Hypotheses 1 & 2

Our analyses tested whether the probability of infection was associated with the reporting of symptoms (hypothesis 1), and whether infection intensity had an effect on symptom reporting (hypothesis 2). Our ability to test hypothesis 1 may

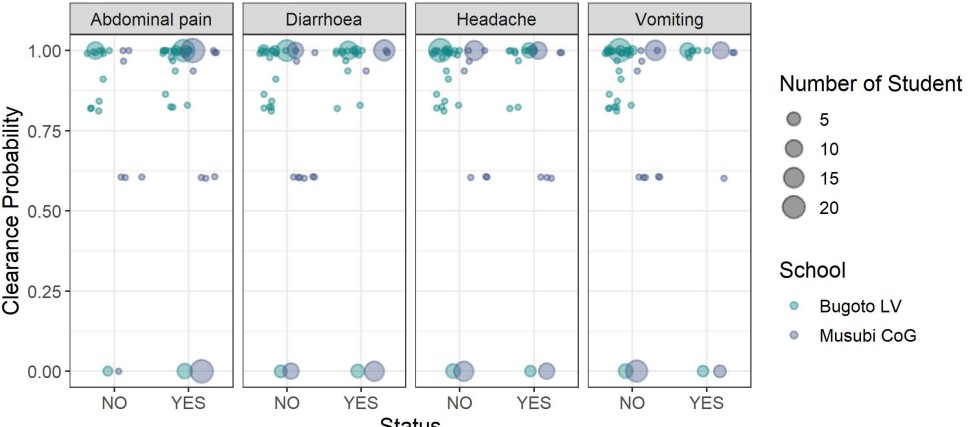

**Fig 4. The clearance probability from the latent class analysis compared with not reporting (left – NO) or reporting (right – YES) abdominal pain, diarrhoea, headache, or vomiting as side effects after praziquantel treatment in Bugoto Lake View (LV) (dark blue) and Musubi Church of God (CoG) (green) primary schools.** The size of dots relates to the number of participants.

**Table 6. Regression results of the beta distribution models of other side effects (abdominal pain, diarrhoea, headache, and vomiting) and clearance probability after praziquantel treatment with the side effect status as independent variable. The intercept was the group of participants who did not report the side effect in school of Bugoto Lake View (LV) and Musubi Church of God (CoG) primary schools respectively. Statistical significance is indicated as follows: \*\*\* p<0.001, \*\* p<0.01, \* p<0.05, and p≥0.05 is considered non-significant.**

| Independent Variable | School | Side Effect | Effect | Estimate±SE | p-value# |
|---|---|---|---|---|---|
| Clearance Probability | | | | | |
| | Bugoto LV | Abdominal pain | (Intercept) | 1.016±0.269 | **< 0.001 (\*\*\*)** |
| | | | Side effect showed | -0.034±0.304 | 0.911 |
| | | Diarrhoea | (Intercept) | 1.036±0.211 | **< 0.001 (\*\*\*)** |
| | | | Side effect showed | -0.098±0.286 | 0.731 |
| | | Headache | (Intercept) | 0.999±0.193 | **< 0.001 (\*\*\*)** |
| | | | Side effect showed | -0.023±0.313 | 0.942 |
| | | Vomiting | (Intercept) | 1.011±0.186 | **< 0.001 (\*\*\*)** |
| | | | Side effect showed | -0.084±0.349 | 0.809 |
| | Musubi CoG | Abdominal pain | (Intercept) | 0.313±0.511 | 0.540 |
| | | | Side effect showed | -0.186±0.545 | 0.732 |
| | | Diarrhoea | (Intercept) | 0.142±0.272 | 0.603 |
| | | | Side effect showed | 0.014±0.360 | 0.969 |
| | | Headache | (Intercept) | 0.114±0.230 | 0.620 |
| | | | Side effect showed | 0.089±0.365 | 0.808 |
| | | Vomiting | (Intercept) | 0.017±0.210 | 0.935 |
| | | | Side effect showed | 0.476±0.397 | 0.230 |

# Significance codes: 0 – 0.001 (\*\*\*); 0.001 – 0.01 (\*\*); 0.01 – 0.05 (\*); >0.05 (non-significance).

have been limited, particularly in Musubi CoG, by the strong imbalance in the data, as over 90% of participants were classified as infected pre-treatment, reducing the statistical power to detect an association. For hypothesis 2, the relationship between symptoms and infection intensity was complex, varied between schools, and was often non-linear. Reports of blood-in-stool did increase with increasing infection intensity in both schools up to a point, after which there was a strong decline in the association.

It is possible that the decline in symptom prevalence at higher egg counts may have been exaggerated by the small sample sizes for individuals with incredibly high egg counts, especially for symptoms that were infrequently reported. However, there is increasing recognition that infection intensity as measured by eggs excreted in stool is not linearly related to morbidity [40,52]. In the cases of nausea and pain-when-urinating at Bugoto LV, and abdominal pain, diarrhoea, nausea and itching/rash at Musubi CoG, prevalence of these symptoms was notable even in those who were classified as not infected (i.e., 0 epg and with negative or low antigen scores and therefore more likely to be estimated as uninfected as by the model and truly uninfected). This suggests that these are not only associated with schistosomiasis infection – and are generally not related to infection intensity, and may indicate that whatever is causing these morbidities is a stronger driver or is at least not exacerbated by schistosomiasis.

Symptoms of infection with soil transmitted helminths (STHs) can also resemble *S. mansoni* morbidities [4,12–15,53], and they are known to often co-occur in the same communities and individuals [42,54]. The possibility of co-infection with *S. mansoni* and STHs might explain some of the symptoms reported prior to treatment where no relationship was found with *S. mansoni* infection or intensity. Bugoto LV also received Albendazole treatment in 2003 when they received their initial MDA, therefore it is possible that STH – *S. mansoni* co-infection was less prevalent, potentially explaining why participants from Bugoto LV reported fewer pre-treatment symptoms. Additionally, regions with a high incidence of schistosomiasis often overlap with areas endemic for malaria, suggesting that malaria should also be considered as a potential contributing factor of symptom severity [55–57], likely impacting the presence and/or reporting of symptoms. Additionally, limited access to sanitation facilities, and reliance on untreated lake water further can lead to high rates of diarrhoeal disease which can also share symptoms albeit commonly much more acute. While these co-infections may explain the lack of relationships found in some cases, our results highlight that specific symptoms, such as blood-in-stool and nausea, were significantly associated with *S. mansoni* infection intensity, underscoring the importance of considering multiple factors when interpreting symptom patterns.

## Hypotheses 3 & 4

Our analyses next tested whether there was a relationship between the probability of infection and the reporting of side effects (hypothesis 3), or infection intensity and the reporting of side effects (hypothesis 4). We found different outcomes for each school. More participants from Musubi CoG reported side effects than those at Bugoto LV. At Bugoto LV, diarrhoea, headache and vomiting increased initially with increasing infection prevalence. Abdominal pain did mildly increase in prevalence at moderate infection intensities but was also prevalent in 50% of uninfected children. The significant negative association observed between abdominal pain and infection intensity may also reflect the low number of individuals in the highest egg count categories, and the high background prevalence of abdominal pain even among uninfected individuals which may influences if someone reports it as a 'new' treatment associated side effect.

The differences of infection intensity and side effects between the two schools could be attributed to the fact that Musubi CoG was praziquantel-naïve in 2004 when these data were collected, while Bugoto LV had already participated in one praziquantel MDA in 2003 [58]. If praziquantel is effective at treating *S. mansoni* infections, and the side effects are a function of worms dying as has been previously suggested, then this may explain our observed differences in side effect reporting. However, since our analysis accounts for infection intensity by comparing participants within the same intensity bins, the overall lower infection intensities in Bugoto LV should not, in itself, drive these differences. If the number of dying worms were the primary driver of side effects, we would expect to see a consistent association between heavier infections and increased side effects across both schools. Despite some students from Bugoto LV still harbouring heavy infections one year after MDA started, we did not observe this pattern [27,41]. One potential explanation for this might be that the worms in children at Bugoto LV had already survived a treatment and could therefore be more resistant to the drug on average and those previously unexposed at Musubi CoG and this warrants further research.

Irrespective of the causes of side effects, simple interventions such as co-administration of medications like paracetamol for headaches and abdominal pain, or antihistamines for itching and rash, could help mitigate effects [14,21,34,35].

Furthermore, given the lack of association, improving health messaging around the lack of a relationship may improve uptake of MDA by reducing stigma associated with potential side effect.

### Hypothesis 5

We tested whether the occurrence of side effects was related to the probability of infection clearance after praziquantel treatment. There was no significant relationship between the reporting of abdominal pain, diarrhoea, or headache as side effects and successful clearance (and the other side effects could not be fitted to the models). This indicates that the presence of these specific side effects does not predict the efficacy of praziquantel treatment in clearing *S. mansoni* infections in these participants and therefore that side effects experienced post-treatment can not be used as reliable indicators of successful worm clearance. This is in contrast to previous studies which suggested that severe side effects might be indicative of substantial worm death, leading to higher treatment efficacy [33,39,59].

### Limitations

One of the limitations of this study is that our surveys recorded symptoms and side effects in a binary manner, which limits the ability to capture detailed variation, and it may be that the severity of specific symptoms does increase with higher infection intensities. To improve upon this, future research could employ scales to provide a more nuanced understanding of symptom and side-effect severity and duration, as well as their impact on health-related quality of life (HRQoL). A recent study highlighted that integrating HRQoL assessments offers valuable insights into the broader effects of treatment on patients' well-being, but demonstrated a complex relationship between infections, symptons and HRQoL [43]. It is also possible that participants who experienced severe symptoms pre-treatment may be less likely to recognise, classify and/or report side effects after treatment, as they already experienced them at the time of treatment and they are not 'new' drug induced side effects. Again, a scale of the severity of symptoms and side effects may help to elucidate if pre-treatment symptoms affect post-treatment side effect reporting. We had hoped to address this question, but upon further investigation, our data were not appropriate to statistically analyse this and a more systematic method of recording pre treatment symptoms and post-treatment side effect data is needed. Without these improved data, here it was not possible to delineate whether a participant had not reported the side effect because it was not there, or because it was not worse than the symptom they may have been experiencing prior to treatment.

Another limitation of our analysis was the small number of participants with very high pre-treatment infection intensities, which is epidemiologically reasonable but resulted in sparse data within higher intensity categories, leading to a strong skew towards zero in the proportion of individuals reporting symptoms or side effects in these high infection intensity categories. These data sparsity may explain why in Fig 2, most symptom and side effect reporting curves decline rapidly to zero at higher infection intensities.

A further limitation is the application of a second treatment to children with >100epg at one-week post-treatment, which could have implications for our results. However, this additional treatment only impacts our findings relevant to hypothesis 5 (the probability of clearance) – it would have no impact on symptom reporting pre-treatment, or side effect reporting as these data were collected after the first treatment. It is possible that in relation to the single treatments commonly provided now, we are over-estimating treatment efficacy. We did not stratify our analysis by single and double treatments due to the small sample sizes that would result within each group and because this would likely be confounded with pre-treatment infection intensity. However, we note that a second treatment did not equal 100% probability of clearance in all children, nor was it strongly associated with symptoms, side effects or overall drug efficacy, as illustrated in S4 Fig and therefore likely did not warrant separate analyses.

We also observed that some participants experienced vomiting after taking the medication, but our current survey did not track whether they continued to experience this side effect after subsequent doses. Future research should

investigate whether vomiting and its timing impact treatment outcomes by including follow-up assessments after drug re-administration.

Finally, these data were also collected around 20 years ago. However, the distribution of infection intensities and infection prevalence within these communities are still similar [43] suggesting the epidemiological context has not substantially changed. The students from Musubi CoG, were drug naïve, so even older children were receiving their first treatment, which is now unlikely given concerted treated efforts. However, there are no differences in the reporting of side effects between the two schools and so no difference between children being exposed to praziquantel for the first or second time, which means these findings are likely still relevant because responses are likely due to overall exposure to praziquantel, not just relevant to the first ever treatment.

## Summary and conclusions

Mass drug administration (MDA) remains the cornerstone of schistosomiasis control and plays a key role in achieving the WHO 2030 targets for EPHP. Although praziquantel is effective against *S. mansoni*, it can cause temporary side effects such as abdominal pain, diarrhoea, and vomiting and challenges remain around how to address concerns about side effects, treatment adherence, and reinfection dynamics. As control programmes progress, managing side effects and improving community engagement with evidence-based information about them may be critical to improving MDA uptake and adherence. However, we found no evidence that side effects or symptoms were positively associated with infection intensities, being infected, or treatment efficacy. Fewer side effects were reported in the school (Bugoto LV) which had had one previous round of treatment, than from Musubi CoG, where treatment was being delivered for the first time. Further research will help understand if such differences are due to different previous praziquantel exposure or other factors and how treatment history may influence the frequency or severity of reported side effects, especially now as control programmes have now been running for many years, which may result in different findings in comparison to this study which analyses data from the start of Uganda's control programme.

## Supporting information

**S1 Table. The number of students in each school participating in the research in 2004.**
(DOCX)

**S1 Fig. The *Schistosoma mansoni* infection intensity of every child in Bugoto Lake View (LV) (top graph in both panels) and Musubi Church of God (CoG) (bottom graph in both panels) primary school who reported (orange), or did not report (blue) pre-treatment symptoms (x-axis of top panels) or post-treatment side effects (x-axis of bottom panels) in 2004.** Note the jittering of points to prevent them overlapping.
(DOCX)

**S2 Fig. The proportion of Bugoto Lake View primary school children who reported (orange), or did not report (blue) pre-treatment symptoms (top row) and post-treatment side effects (bottom row) compared with *Schistosoma mansoni* infection probability.**
(DOCX)

**S3 Fig. The proportion of Musubi Church of God primary school children who reported (orange), or did not report (blue) pre-treatment symptoms (top row) and post-treatment side effects (bottom row) compared with *Schistosoma mansoni* infection probability.**
(DOCX)

**S4 Fig. The clearance probability compared with status of other self-reported side effects (abdominal pain, diarrhoea, headache, and vomiting) after single (dark purple) or double (dark green) praziquantel treatment in Bugoto Lake View and Musubi Church of God primary schools, in 2004.** The size of dots means number of students.
(DOCX)

## Acknowledgments

We are sincerely grateful to the National and District officials, and drivers from the Vector Control Division of the Ministry of Health for their support. We also thank head teachers and teachers from Bugoto LV and Musubi CoG for their assistance in the field. Our appreciations go to the village health teams and village local council chairpersons for their efforts in mobilizing the community to participate in this study. Lastly, we are deeply grateful to the participants who provided samples and data for this research.

## Author contributions

**Conceptualization:** Joanne P. Webster, Poppy H. L. Lamberton, Jessica Clark.

**Data curation:** Moses Arinaitwe, Adriko Moses, Narcis Kabatereine, Edridah M. Tukahebwa, David W. Oguttu, Aidah Wamboko, Annet Namukuta, Annet Enzaru, Alan Fenwick, Joanne P. Webster, Poppy H. L. Lamberton, Jessica Clark.

**Formal analysis:** Huanghehui Yu, Joaquin M. Prada, Jessica Clark.

**Funding acquisition:** Alan Fenwick, Joanne P. Webster, Poppy H. L. Lamberton.

**Investigation:** Huanghehui Yu, Joaquin M. Prada, Poppy H. L. Lamberton, Jessica Clark.

**Methodology:** Joaquin M. Prada, Joanne P. Webster, Poppy H. L. Lamberton, Jessica Clark.

**Project administration:** Narcis Kabatereine, Edridah M. Tukahebwa, Alan Fenwick, Joanne P. Webster, Poppy H. L. Lamberton.

**Software:** Huanghehui Yu, Joaquin M. Prada, Jessica Clark.

**Supervision:** Huanghehui Yu, Poppy H. L. Lamberton, Jessica Clark.

**Validation:** Poppy H. L. Lamberton, Jessica Clark.

**Visualization:** Huanghehui Yu, Jessica Clark.

**Writing – original draft:** Huanghehui Yu, Poppy H. L. Lamberton, Jessica Clark.

**Writing – review & editing:** Huanghehui Yu, Moses Arinaitwe, Adriko Moses, Narcis Kabatereine, Edridah M. Tukahebwa, David W. Oguttu, Aidah Wamboko, Annet Namukuta, Annet Enzaru, Joaquin M. Prada, Joanne P. Webster, Poppy H. L. Lamberton, Jessica Clark.

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
