## [Decision Letter · Decision Letter 0]

1 Jul 2025

The associations between Schistosoma mansoni infection, pre-treatment symptoms, praziquantel side effects, and treatment efficacy in Ugandan school-aged children

Dear Dr. Clark,

Thank you for submitting your manuscript to PLOS Neglected Tropical Diseases. After careful consideration, we feel that it has merit but does not fully meet PLOS Neglected Tropical Diseases's publication criteria as it currently stands. Therefore, we invite you to submit a revised version of the manuscript that addresses the points raised during the review process.

Please submit your revised manuscript within 60 days Aug 30 2025 11:59PM. If you will need more time than this to complete your revisions, please reply to this message or contact the journal office at plosntds@plos.org. Please include the following items when submitting your revised manuscript:

We look forward to receiving your revised manuscript.

Kind regards,

David J. Diemert, M.D.

Academic Editor

Eva Clark

Section Editor

Shaden Kamhawi

co-Editor-in-Chief

Paul Brindley

co-Editor-in-Chief

**Journal Requirements:**

At this stage, the following Authors/Authors require contributions: Huanghehui Yu, Moses Arinaitwe, Adriko Moses, Narcis Kabatereine, Edridah M. Tukahebwa, David W. Oguttu, Aidah Wamboko, Annet Namukuta, Annet Enzaru, Joaquin M. Prada, Alan Fenwick, Joanne P. Webster, Poppy H. L. Lamberton, and Jessica Clark. Please ensure that the full contributions of each author are acknowledged in the "Add/Edit/Remove Authors" section of our submission form.

4) We have noticed that you cited Figure S5 on page 11 line 236 in your manuscript. However, there is no corresponding file uploaded to the submission. Please upload it as a separate file with the item type 'Supporting Information'. Or, if the figure is no longer to be included as part of the submission please remove all reference to it within the text.

5) We have noticed that you have uploaded Supporting Information files, but you have not included a list of legends. Please add a full list of legends for your Supporting Information files after the references list.

2) If any authors received a salary from any of your funders, please state which authors and which funders.

7) Please ensure that the funders and grant numbers match between the Financial Disclosure field and the Funding Information tab in your submission form. Note that the funders must be provided in the same order in both places as well. Currently, the order of the funders is different in both places. In addition, this grant "EPX0270821" is missing from the Financial Disclosure field.

**Reviewers' Comments:**

Reviewer's Responses to Questions

**Key Review Criteria Required for Acceptance?**

**Methods**

-Are the objectives of the study clearly articulated with a clear testable hypothesis stated?

-Is the study design appropriate to address the stated objectives?

-Is the population clearly described and appropriate for the hypothesis being tested?

-Is the sample size sufficient to ensure adequate power to address the hypothesis being tested?

-Were correct statistical analysis used to support conclusions?

-Are there concerns about ethical or regulatory requirements being met?

Reviewer #1: Line 170-172, Please add the batch number of the POC-CCA test that was used. Was POC-CCA testing done on fresh urines in 2004, or retrospectively (after storage)?

Reviewer #2: •Line 157: what was the intended sample size and calculation? The actual number sampled and why the sample size was not met should be covered in the Results section.

•Line 162: If not included in analysis then don’t include in the methods

•Line 170: Was urine collected and tested for POC-CCA in these sites 2004?

•Line 172: This implies the sample children received PZQ following the study as well as PZQ through the national MDA programme (lines 150 – 152)?

•Line 177: Participants with > 100 epg at 1 week received a second praziquantel dose – there is no mention of how this alters the four-week clearance probabilities and whether analyses stratified by single vs. double dosing are feasible. The Limitations section downplays the importance of this key factor which is likely significantly skewing the results of the study (line 548)

•Line 182: Need to say why the one-week post-treatment data not included.

•Line 192: S. haematobium

•Line 219: Trace as positive or negative?

**Results**

-Does the analysis presented match the analysis plan?

-Are the results clearly and completely presented?

-Are the figures (Tables, Images) of sufficient quality for clarity?

Reviewer #1: Line 294, Table 1:

- Please adjust the decimals, now some percentages do not have decimals while others have 2 decimals (1 decimal would be sufficient). The same for the text (in line 284 no decimal, while in the subsequent paragraph 2 decimals are used).

- Can you specify whether the symptoms pre-treatment were ongoing (if they had them presently) or if they had ever experienced them? (same question for the text in line 286-289).

Line 315-316, In Table 2 you also include headache as symptom pre-treatment, but this is not mentioned in the text?

Line 314-327, please be clear in the text whether you talk about pre-treatment symptoms or post-treatment side effects. For example in line 320-323 it is not clear if this is symptoms or side effects (while in the next sentence you specific by referring to the pink lines in the figure).

Reviewer #2: •Line 281: we know this from the Methodology section

•Line 282: we have no idea whether the number of children sampled is sufficient to perform the analysis. Why were there nearly twice as many participants for Bugoto than for Musubi?

•Line 282: These are not final numbers in the S1 Table

•Line 284: what are these values? Assume eggs per gram?

•Line 286-287: This are not the same figures as in Table 1?

•Line 290: Figures are not same as in table.

Table 1- Please can the authors review and ensure everything in the table is explained i.e. what are the percentages?; where are the POC-CCA results?

Tables / Figures - too many – which are the key ones

**Conclusions**

-Are the conclusions supported by the data presented?

-Are the limitations of analysis clearly described?

-Do the authors discuss how these data can be helpful to advance our understanding of the topic under study?

-Is public health relevance addressed?

Reviewer #1: The authors conclude that co-infections and other sources of morbidity may influence the reporting of symptoms and side effects. Could the authors clarify how this conclusion was drawn, given that other infections or causes of morbidity do not appear to have been directly assessed in the study?

Reviewer #2: •The conclusions are supported by the data presented however statements about symptoms resolving at lower intensity thresholds overstate what your four‐week follow-up can demonstrate, given the sparse data in the highest‐intensity categories

•Limitation - small sample sizes at high intensity levels and the binary nature of symptom reporting are acknowledged but there also needs to be mention of potential semi-quantitative misclassification in the POC-CCA scoring (“Trace” vs. “+”) and the absence of longer-term (beyond four weeks) efficacy data.

•A brief caveat about unmeasured confounders (co-infections, immunity status) should be included in the limitations.

**Editorial and Data Presentation Modifications?**

Reviewer #1: (No Response)

Reviewer #2: (No Response)

**Summary and General Comments**

Reviewer #1: This manuscript addresses a critical barrier to effective schistosomiasis control: the role of treatment side effects in MDA uptake, particularly among heavily infected individuals. A better understanding of the relationship between side effects, infection intensity, and treatment efficacy is essential to improve community participation and maximize the impact of control programs. The data presented in this manuscript is very extensive, but clearly described and discussed for each hypothesis.

However, I have some questions:

Since the study focused on school-aged children, I was curious to what extent children actually refuse praziquantel during MDA. The study from Adriko et al (reference 38) concludes that being enrolled in a school actually improves the odds of taking praziquantel. Is the study population in your study representative, or should praziquantel uptake also be investigated in adults? Is treatment refusal more commonly seen among adults in the community?

While the study offers interesting insights, I am concerned about the relevance of the findings given that the data were collected over 20 years ago (2004). The manuscript does not discuss how the epidemiological context may have changed since then, particularly in terms of prevalence, intensity of infection, and overall health status of affected populations. The authors should address this limitation in the discussion, and explain how their findings can be interpreted in light of current MDA practices and transmission dynamics, and to discuss the limitations of applying results from such historical data to today's settings.

How accurate are the high(er) egg counts, eg above 500 EPG? It would be good to also mention the infection intensity categories based on KK. For example the mean EPG Musubi CoG is >400 indicating heavy infection intensity, while the mean EPG for Bugoto is ~100 indicating moderate infection intensity. Since the infection intensity (EPG) is most likely not normally distributed, I would be more interested in the median EPG and the range. Is the median EPG per school showing the same?

The study uses pre-treatment reported symptoms in the analysis, but in the results the authors do not specify whether these pre-treatment symptoms are present at the moment or if they had experienced these symptoms in the past. I would be more interested to see the analysis focused on ongoing symptoms and whether this was associated with infection probability.

Reviewer #2: The manuscript needs to be re-read and significantly edited by the first and senior authors to ensure it meets publishing quality, there are quite a few inconsistencies in terms of side effects and side-effects; minor mis-spellings (there are a lot of extra ‘s’ on words), consistent order of narrative e.g side-effects and symptoms vs symptoms and side-effects, the names of the schools; overuse of the word hypotheses/hypothesises etc. The manuscript comes across as a student project dissertation rather than a publication ready for peer-review.

The manuscript is relatively long and the methodology, results and discussion sections are overly verbose and need a major rewrite to be made more succinct. There is also no mention of the sample size and how it was calculated.

There are many occasions where the numbers in narrative don’t match those in the tables/figures and information also missing.

There are specific comments for each section up until Results where it became clear a major revision of the manuscript was required as the narrative and tables/figures didnt share the same information.

Specific comments for other sections

Abstract

•Line 24: use words to describe ‘heavily/infected’ people – clarify if this means infected people and heavily infected people or heavily infected people?

•Lines 33 – 39: Principal Findings – I read this a few times and wasn’t clear on what it was trying to communicate and how it linked to the Conclusion section. Rewrite to be clearer.

•Lines 43-44: its not clear what would be supporting informed discussions with communities or what is inferred by that. Suggest removing or making clearer.

Author summary

•Line 55: ‘discomforts’ is very subjective – change to be more specific

•Line 58: It is unclear what has been said previously will improive community engagement? There is a large leap between resercah studies determining something and community engagement – need to be clearer.

Introduction

•Line 60: change to elimination as a public health problem goals

•Line 69: remove ‘warm’ or replace with specific temperature

•Line 73: inaccessible by what/whom? Directly unobservable by what/whom? Re-write to be clearer.

•Line 80: Kato-Katz

•Line 80 – 81: Latent class analyses haven’t been developed specifically for schistosomiasis diagnostics – sentence needs reworded to be clearer.

•Line 88: ‘This’ what? – need to be clear on what ‘This’ is referring to.

•Line 107, 108, 114: all sentences start very similarly, need to edit to introduces the sentences differently

•Line 118: presence or prevalence?

•Line 138 - 140: #5 has not been addressed in the Intro – why do the authors hypothesise this?

PLOS authors have the option to publish the peer review history of their article (what does this mean? ). If published, this will include your full peer review and any attached files.

**Do you want your identity to be public for this peer review?** For information about this choice, including consent withdrawal, please see our Privacy Policy .

Reviewer #1: **Yes: ** PT Hoekstra

Reviewer #2: No

**Figure resubmission:**

**Reproducibility:**



---

## [Decision Letter · Decision Letter 1]

1 Oct 2025

Dear Dr Clark,

We are pleased to inform you that your manuscript 'The associations between Schistosoma mansoni infection, pre-treatment symptoms, praziquantel side effects, and treatment efficacy in Ugandan school-aged children' has been provisionally accepted for publication in PLOS Neglected Tropical Diseases.

Best regards,

David J. Diemert, M.D.

Academic Editor

Eva Clark

Section Editor

Shaden Kamhawi

co-Editor-in-Chief

Paul Brindley

co-Editor-in-Chief

Reviewer's Responses to Questions

**Key Review Criteria Required for Acceptance?**

**Methods**

-Are the objectives of the study clearly articulated with a clear testable hypothesis stated?

-Is the study design appropriate to address the stated objectives?

-Is the population clearly described and appropriate for the hypothesis being tested?

-Is the sample size sufficient to ensure adequate power to address the hypothesis being tested?

-Were correct statistical analysis used to support conclusions?

-Are there concerns about ethical or regulatory requirements being met?

Reviewer #1: (No Response)

**Results**

-Does the analysis presented match the analysis plan?

-Are the results clearly and completely presented?

-Are the figures (Tables, Images) of sufficient quality for clarity?

Reviewer #1: (No Response)

**Conclusions**

-Are the conclusions supported by the data presented?

-Are the limitations of analysis clearly described?

-Do the authors discuss how these data can be helpful to advance our understanding of the topic under study?

-Is public health relevance addressed?

Reviewer #1: (No Response)

**Editorial and Data Presentation Modifications?**

Reviewer #1: (No Response)

**Summary and General Comments**

Reviewer #1: The authors have adequately addressed all my comments and provided clear justifications where necessary. The revisions have substantially improved the manuscript, and I am satisfied with the changes made. I have no further questions and recommend to accept the manuscript.

PLOS authors have the option to publish the peer review history of their article (what does this mean? ). If published, this will include your full peer review and any attached files.

**Do you want your identity to be public for this peer review?** For information about this choice, including consent withdrawal, please see our Privacy Policy .

Reviewer #1: **Yes: ** PT Hoekstra

---

## [Editor Report · Acceptance letter]

Dear Dr Clark,

We are delighted to inform you that your manuscript, "The associations between Schistosoma mansoni infection, pre-treatment symptoms, praziquantel side effects, and treatment efficacy in Ugandan school-aged children," has been formally accepted for publication in PLOS Neglected Tropical Diseases.

Best regards,

Shaden Kamhawi

co-Editor-in-Chief

Paul Brindley

co-Editor-in-Chief
